# MAN5, a Glycosyl Hydrolase Superfamily Protein, Is a Key Factor Involved in Cyanide-Promoted Seed Germination in *Arabidopsis thaliana*

**DOI:** 10.3390/genes14071361

**Published:** 2023-06-27

**Authors:** Lu-Lu Yu, Fei Xu

**Affiliations:** 1Applied Biotechnology Center, Wuhan University of Bioengineering, Wuhan 430415, China; lulu2019@whsw.edu.cn; 2College of Life Science and Technology, Wuhan University of Bioengineering, Wuhan 430415, China

**Keywords:** mannosidase, cyanide, seed germination, stress

## Abstract

Seed germination is the complex adaptive trait of higher plants influenced by a large number of genes and environmental factors. Numerous studies have been performed to better understand how germination is controlled by various environmental factors and applied chemicals, such as cyanide. However, still very little is known about the molecular mechanisms of how extrinsic signals regulate seed germination. Our and previous studies found that non-lethal cyanide treatment promotes seed germination, but the regulatory mechanism is unclear. In this study, we found that a low concentration of cyanide pretreatment significantly enhanced the expression of endo-β-mannanase 5 (*MAN5*) gene in *Arabidopsis thaliana*, and the mutation of this gene impaired cyanide-mediated seed germination. In contrast, overexpression of *MAN5* gene enhanced Arabidopsis seed germination ability under both normal and salt stress conditions. Further studies showed that the expression of the *MAN5* gene was negatively regulated by ABA insensitive 5 (ABI5); In *abi5* mutant seeds, the expression of the *MAN5* gene was increased and the seed germination rate was accelerated. Additionally, cyanide pretreatment markedly reduced the gene expression of *ABI5* in Arabidopsis seeds. Taken together, our data support the involvement of *MAN5* as a key gene in cyanide-mediated seed germination and confirm the role of ABI5 as a critical negative factor involved in cyanide-regulated *MAN5* gene expression.

## 1. Introduction

Seed germination is affected by a series of physiological changes, gene expression regulation, protein modification, and coordination of metabolic responses [1], and there are many unsolved mysteries. In the past decades, studies on seed germination mostly focuses on intracellular plant hormone level changes and their regulation [2,3]. Abscisic acid (ABA) and gibberellin (GAs) have antagonistic relationships in regulating seed dormancy and germination, with the former inducing dormancy and the latter promoting seed germination [4]. In fact, in addition to plant hormones, some small molecules have also been shown to play key regulatory roles in seed germination; For example, nitric oxide (NO) and reactive oxygen species (ROS) [5,6,7].

Notably, as a small molecule compound, cyanide (e.g., hydrogen cyanide, HCN) has long been known as a toxic substance because it binds to cytochrome *c* oxidase and inhibits cellular respiration in mitochondria. However, cyanide can be produced by plant intracellular metabolism, one of which is a by-product of ethylene synthesis [8]. In addition, cyanide can also be produced by hydrolysis of cyanogenic glycosides and cyanolipids [9]. Interestingly, during the combustion of plants, pyrotechnics were found to contain small gas molecules—HCN [10]. The burned grass ash also contains cyanol compounds, which can release HCN and enter the soil after hydrolysis in water [11].

Recently, cumulative studies have confirmed that cyanide may not be just a toxic waste, it may have beneficial effects at low concentrations. Some studies suggest that cyanide plays significant roles in enhancing plant stress acclimation [12] and seed germination [13,14]. Our previous studies also found that low-concentration cyanide pretreatment enhanced tomato seed germination [15]. However, how cyanide affects seed germination remains unclear in terms of physiological, biochemical, metabolic levels, and regulation of gene expression. It is worth noting that there has been a lack of systematic research on cyanide in the past, especially cyanide as a signaling molecule involved in seed germination regulation, and it still faces insufficient evidence and little research. What are the key genes for cyanide regulation, and does it function as a signaling molecule? This is a key issue that needs to be addressed to explain the beneficial aspects of cyanide.

Our previous work showed that low concentrations of cyanide can significantly induce tomato seed germination and regulate the expression of many genes, among which the mannanase genes were markedly upregulated; e.g., the gene expression of 1,4-β-mannosidase 2 (MAN2; also named as MAN5) was upregulated by about 8500 (log_2_FC = 12.83) [15]. The endo-β-mannanases (MANs; EC. 3.2.1.78) family consists of hydrolytic enzymes that catalyze the cleavage of β-1,4-linkage in the mannan backbone, thereby loosening the cell wall and releasing the mannose units [16]. Studies have shown that mannose helps strengthen plant resistance to environmental stresses [17,18]. Importantly, some members of *MAN* family genes have been confirmed to participate in the softening process of the endosperm wall and helps promote the breakthrough of the radicle through the seed coat [19,20], suggesting that they may be “new key genes” in the process of cyanide-induced seed germination. Nevertheless, how cyanide regulates the gene expression of *MANs* and the underlying mechanism remains to be further studied and elucidated. Therefore, this study aimed to explore the effects of cyanide on the expression of *MAN* family genes in Arabidopsis, in which the role of *AtMAN5* gene expression in cyanide-mediated seed germination was emphasized.

## 2. Materials and Methods

### 2.1. Plant Material and Growth Conditions

Seeds of Arabidopsis including wild-type (WT; Columbia-0, Col-0), *MAN5* gene overexpression and T-DNA insertion mutants, and *ABI5* gene mutants were used in the present study. Seeds were sourced as follows: The Arabidopsis Col-0, *MAN5* gene T-DNA insertion mutant seeds (NASC ID: N664445 and N675644) and *ABI5* gene T-DNA insertion mutant seeds (NASC ID: N688157) were obtained from Nottingham Arabidopsis Stock Centre (NASC). In this study, primers were used for T-DNA verification from SALK (http://signal.salk.edu/tdnaprimers.2.html, accessed on 10 May 2023) to identify the correct seeds for subsequent experimental studies. The plants are grown in greenhouses with 16 h of light (approx. 120 μmol m^–2^ s^–1^) at 22 °C and 8 h of the dark at 18 °C, 70% relative humidity.

To generate the *MAN5* gene overexpressing plants, cDNA fragments of At4g28320 including ORF sequence were synthesized by BT lab (Bio-Transduction Lab Co., Ltd, Wuhan, China) and cloned into pCambia1300 vectors carrying 35S promoter. Plants overexpressing *MAN5* genes were transformed by the floral-dip method [21], and transgenic lines were selected on media containing 80 µM kanamycin (Sigma, St. Louis, MO, USA). Seeds of the T_2_ generation were used throughout the study. The relative gene expression of *MAN5* overexpressing plants was shown in Appendix A.

### 2.2. Chemical Treatment

For cyanide treatment, seeds from the WT, *MAN5* gene overexpression and T-DNA insertion mutants were stratified with ddH_2_O for 24 h at 4 °C in the dark. Then, all seeds were surface-sterilized with 8% NaClO for 10 min, followed by six washes with sterile distilled water. After that, some seeds were pretreated with 10 μM HCN (This is the optimal treatment concentration based on our previous research results [15]) for 2~24 h at 25 °C in a sealed glass container. After pretreatment, cyanide was removed and the seeds were placed onto 1/2 MS agar plates for germination. All treatments had three biological replicates in each experiment, and three or more independent experiments were carried out for the whole study. In these experiments, three biological replicates are seeds harvested in three batches of the same species. At least 200 seeds were used for each replicate experiment.

To further compare the differences in seed germination and growth under normal and salt stress conditions, the above seeds were sterilized with 8% NaClO and rinsed five times with ddH_2_O (control) or 100 mM NaCl solution. After that, seeds were plated onto 1/2 MS medium containing 100 mM NaCl solution.

### 2.3. Germination Assay

The germination of seeds was counted over time, and a seed was considered germinated when the radicle protruded 1~2 mm. The germination percentage (%) and germination rate (T_50_) were recorded and compared between different samples under normal and salt stress conditions. T_50_, time to obtain 50% of germinated seeds.

### 2.4. Measurement of Total Chlorophyll Contents

The total chlorophyll of Arabidopsis seedlings was extracted and measured according to previously described methods [22]. Approximately 0.5 g of samples were ground with 5 mL of 80% acetone. Following centrifugation at 10,000× *g* for 15 min, at 4 °C, 1 mL of supernatant was used to read the absorbance values of chlorophyll *a* and chlorophyll *b* at the wavelength of 663 and 645 nm, respectively, using a spectrophotometer (TU1800 spectrophotometer, P-general Limited Company, Beijing, China). Total chlorophyll contents were calculated and expressed as mg per gram of fresh weight.

### 2.5. HPLC for ABA and GA Determination

The levels of hormone including ABA and GA_3_ were measured by HPLC according to the previous work [15]. For GA_3_ extraction and measurement, seeds were ground carefully with liquid nitrogen, and 0.2 g of the powdered sample was extracted overnight at 4 °C with 1.5 mL 70% (*v*/*v*) acetonitrile. After vortexing for 30 s and centrifugation at 14,000 rpm for 10 min, 1.0 mL supernatant was collected and then evaporated to dryness under nitrogen gas stream at room temperature, constituted in 100 mL of 80% (*v*/*v*) methanol, diluted to 800 mL with water. The extracts were passed through the SPE cartridge (CNWBOND Carbon-GCB SPE Cartridge, 200 mg, 3 mL; Anpel, Shanghai, China) and evaporated to dryness under nitrogen gas stream at room temperature. Next, the samples were reconstituted in 200 mL of 80% (*v*/*v*) methanol and filtered (PTFE, 0.22 μm; Anpel, Shanghai, China) before LC–MS/MS analysis.

For ABA extraction and measurement, 50 mg samples were ground with liquid nitrogen and extracted with 0.5 mL of methanol/water/formic acid (15:4:4, *v*/*v*/*v*) at 4 °C. The extract was vortexed for 10 min and centrifuged at 14,000 rpm for 5 min at 4 °C. The supernatants were collected and the steps above were repeated. The combined extracts were evaporated to dryness under nitrogen gas stream, reconstituted in 80% methanol (*v*/*v*), ultraphoniced (1 min), and filtered (PTFE, 0.22 μm; Anpel, Shanghai, China) before LC-MS/MS analysis.

All of the standards were purchased from Olchemim Ltd. (Olomouc, Czech Republic) and Sigma (St. Louis, MO, USA).

### 2.6. Bioinformation Analysis

(1)Analysis of the cis-acting element of the *MAN5* gene promoter

The 1500-bp sequence upstream of the transcription start site of *MAN5* gene was downloaded from the NCBI website (https://www.ncbi.nlm.nih.gov/, accessed on 10 May 2023), and cis-acting elements in the promoter region were identified using the PLACE online tool (https://www.dna.affrc.go.jp/PLACE/, accessed on 10 May 2023) [23].

(2)Analysis of chromosomal location

The chromosome locations of the Arabidopsis *MAN* family genes were generated by TBtools [24], and the Arabidopsis GFF3 file was downloaded from NCBI (https://www.ncbi.nlm.nih.gov/genome?term=chr17&cmd=DetailsSearch, accessed on 10 May 2023).

### 2.7. Total RNA Extraction and Quantitative Real Time PCR (qRT-PCR)

Total RNA was extracted from different samples (100 mg) using the Trizol reagent (Invitrogen) and following the manufacturer’s instructions. The first-strand cDNA synthesis was reverse transcribed from DNase I-treated RNA with oligo (dT)18 as the primer. qRT-PCR reactions were performed with the ChamQ SYBR qPCR Master Mix (Vazyme Biotech Co., Ltd., Nanjing, China) in a ABI7500 cycler (Applied Biosystems) with three technical repeats for each sample. Reactions were initiated at 94 °C for 15 min followed by 40 cycles at 94 °C for 30 s, 55 °C for 30 s, and 72 °C for 30 s. The relative quantitation of the target gene expression level was performed using the comparative Ct (threshold cycle) method. The amplification of the *PP2AA3* gene (encoding protein phosphatase 2A subunit A3, At1g13320) was used for an internal control. Primers used for qRT-PCR are listed in Appendix A.

### 2.8. Statistical Analysis

The statistical analysis of the results from three independent experiments with nine measurements used a one-way ANOVA, followed by Tukey’s HSD post hoc test or Dunnett’s HSD test. Asterisks or different lowercase letters in graphs indicate the level of significance difference (*p* < 0.05).

## 3. Results

### 3.1. MAN5 Gene Expression Is Markedly Induced by External Cyanide Pretreatment

In this study, the chromosomal mapping of the *MAN* family genes was analyzed and shown in Figure 1A. The results showed that there are three genes located on one of the chromosomes, the *MAN1* gene is located on chromosome 1, the *MAN2* gene is located on chromosome 2, and the *MAN5* gene is located on chromosome 4. The *MAN3* and *MAN4* genes are located on the same chromosome (i.e., chromosome 3) and not far apart; likewise, the *MAN6* and *MAN7* genes are also located on the same chromosome (i.e., chromosome 5), but at opposite ends of the chromosome (Figure 1A). To further investigate the induction of *MANs* gene by cyanide, the gene expression of *MAN1-7* was quantified by qRT-PCR after pretreating with cyanide (10 µmol/L HCN) for 24 h. The results showed that *MAN5* gene had the strongest response to cyanide induction, and its expression was higher than that of other *MAN* family genes after 24 h of cyanide pretreatment, although cyanide also significantly induced the expression of *MAN2*, *MAN6*, and *MAN7* genes (Figure 1B). In comparison, *MAN1*, *MAN3*, and *MAN4* had no significant changes in gene expression with or without cyanide pretreatment (Figure 1B). Therefore, *MAN5* gene overexpression and mutants were selected for all subsequent experiments.

### 3.2. MAN5 Is Essential for Cyanide-Induced Seed Germination

To further explore the effects of changes in *MAN5* gene expression on seed germination, wild-type (WT), *MAN5* gene overexpression (*MAN5*-OE-3, *MAN5*-OE-8) and mutants (N664445 and N675644) seeds were used for germination comparison. As shown in Figure 2, cyanide pretreatment for 12 h helps to enhance the germination rate of WT seeds. In comparison, the germination rate of *MAN5*-OE seeds was significantly higher than that of WT seeds, either in or without cyanide pretreatment (Figure 2C). Importantly, it should be noted that cyanide pretreatment further enhanced the germination speed of *MAN5*-OE seeds (Figure 2B,C). In contrast to overexpressed seeds, the germination rate of *MAN5* mutant seeds was significantly lower than that of WT seeds under normal conditions (Figure 2A,C). When cyanide was applied, the germination speed (T_50_) of *MAN5* mutant seeds was obviously enhanced, but it was still significantly lower than that of WT under the same cyanide pretreatment conditions (Figure 2B,C). Together, these results suggest that MAN5 plays an important role in cyanide-mediated Arabidopsis seed germination.

### 3.3. Overexpression of the MAN5 Gene Helps to Enhance Salt Stress Resistance in Seeds

Next, we analyzed the effects of MAN5 gene overexpression and mutation on seed germination under the conditions of salt stress. As shown in Figure 3, the results showed that seeds overexpressing *MAN5* gene had higher germination speed (T_50_) than WT seeds under 100 mM NaCl stress, although the difference in total germination rate between the two was not significant (Figure 3A,B). In contrast, the germination ability of *MAN5* gene mutant seeds (N664446 and N675644) was significantly inhibited by salt stress, where the germination speed (T_50_) and total germination rate were the weakest compared to WT and *MAN5*-OE seeds (Figure 3A,B). Additionally, comparing the growth after germination, it was shown that the seedlings overexpressed with *MAN5* gene had the best growth, and the fresh weight and chlorophyll content were significantly higher than those of the WT, although the salt pressure forced the leaves of WT and MAN5 seedlings to show different degrees of yellowing (Figure 3C–E). Likely, the growth of *MAN5* gene mutants was worst compared to that of WT and *MAN5*-OE under salt stress conditions; it is worth noting that the cotyledons of the *MAN5* gene mutant did not expand during the same culture period (Figure 3C–E).

### 3.4. Overexpression of the MAN5 Gene Exhibited a Lower ABA/GA_3_ Reatio during Germination

As previous studies have established that phytohormones, especially abscisic acid (ABA) and gibberellin (GA), are the major endogenous factors that act antagonistically in the regulation of seed germination; ABA positively regulates the induction and maintenance of dormancy, whereas GA breaks dormancy and promotes germination [25]. In this regard, changes in the balance of seed ABA/GA levels constitute a central regulatory mechanism underlying the maintenance and release of seed dormancy. In addition, GA_3_ is the most active component of GAs involved in the regulation of seed germination, we next detected and compared the levels of ABA and GA_3_ during germination of WT, *MAN5*-OE, and mutant seeds. The results showed that the relative content of ABA in WT seeds was significantly higher than that of *MAN5*-OE seeds and lower than that of *MAN5* mutant seeds (Figure 4A). Conversely, GA_3_ content accumulates more rapidly in *MAN5*-OE seeds than WT seeds, resulting in a lower ABA/GA_3_ ratio during germination (Figure 4B,C). It is interesting to note that cyanide pretreatment obviously reduced the accumulation of ABA and increase the synthesis of GA_3_, especially during the germination of *MAN5*-OE seeds (Figure 4A,B). In *MAN5* mutant seeds, GA_3_ synthesis was also induced by cyanide, but the ABA/GA_3_ ratio was still higher than that of WT and *MAN5*-OE seeds (Figure 4A–C).

### 3.5. ABI5 Is Involved in Cyanide-Mediated MAN5 Gene Expression Regulation

To date, the most well studied cis-acting elements for endosperm-specific gene expression is the bipartite endosperm box (the E box), which contains the GCN4-like motif targeted by basic leucine zipper (bZIP) transcription factors [26]. To explore the regulatory effect of the *MAN5* gene by cyanide, we then analyzed the cis-acting element in the *MAN5* gene promoter (Appendix A). The results showed that there was one GCN4-like motif (–GAGTCA–) and one ABA-responsive element (–ACGT–) regulated by bZIP transcription factors in the promoter of the *MAN5* gene (Figure 5A). Interestingly, based on our previous transcriptome detection data in tomato seeds, the gene expression of bZIP transcription factor *ABI5* was significantly regulated by cyanide pretreatment [15]. To explore whether *ABI5* is a key gene for regulating *MAN5* in Arabidopsis, changes in *ABI5* gene expression were detected at different times after cyanide pretreatment. The results showed that cyanide pretreatment resulted in significant repression of the expression of *ABI5* gene after 12 h of incubation (Figure 5B). Therefore, we then analyzed the effects of knockout *ABI5* gene on cyanide-induced seed germination and *MAN5* gene expression. Notably, results showed that cyanide-promoted seed germination and induction of the *MAN5* gene were significantly enhanced in the *abi5* mutant (Figure 5C,D), suggesting that the bZIP transcription factor ABI5, as a negative regulator, is involved in cyanide regulation of *MAN5* gene expression.

## 4. Discussion

Although cyanide has been shown to have a positive effect on seed germination at lower concentrations for more than half a century, the molecular mechanism of its action remains largely unknown. In this work, the contribution of *MAN* family genes to the cyanide-induced germination process in Arabidopsis has been explored. Of note, our data demonstrate that the gene expression of *MAN5* was strongly induced by cyanide pretreatment. In addition, overexpression of *MAN5* gene obviously promoted the seed germination of Arabidopsis and enhanced the effect of cyanide, while the mutation of this gene showed the opposite results, compared with WT. These findings indicate that MAN5 might function as a crucial factor involved in cyanide-mediated seed germination in Arabidopsis.

The importance of Arabidopsis MAN5, as a member of the endo-β-mannanase family, in seed germination has been highlighted in previous studies, despite poor information available on its association with other intracellular signaling molecules. In Arabidopsis, a previous study showed that the expression of *MAN2*, *MAN5*, *MAN6*, and *MAN7* genes were detected in dry seeds and induced upon germination [27]. Among them, *MAN5* was confirmed to be the most abundant *MAN* transcript in dry seeds, although *MAN6* knockout had the greatest effect on seed germination [27]. Moreover, it is worth noting that the induced amount of *MAN5* gene after cyanide pretreatment was greater than that of *MAN6* gene in our current study (Figure 1). In tomato, *LeMAN5* (also known as *LeMAN2*), has been proved to be expressed specifically in the micropylar endosperm prior to radicle emergence and its activity increases markedly in the remaining lateral endosperm following radicle protrusion [27,28]. Similar to in tomatoes, transcriptional accumulation of the *MAN5* gene in Arabidopsis was observed specifically in the micropylar endosperm and radicle, and this expression disappeared soon after radicle emergence [27]. It is evident that the endosperm weakening is one of the important processes for radicle emergence in endospermic seeds [28]. Therefore, our findings, together with previous studies, support the idea that *MAN5* is a critical gene involved in Arabidopsis seed germination. Our work found that knocking out *MAN5* not only suppressed the germination rate of Arabidopsis seed, but also significantly weakened cyanide-induced germination, suggesting the importance of *MAN5* in cyanide-regulated seed germination pathways.

In this study, the overexpression of *MAN5* not only contributed to seed germination, but also helped seed germination under moderate saline conditions such as 100 mM NaCl. As a major abiotic stress factor, soil salinity hinders seed germination and subsequent plant growth, development, and yield [29,30]. Therefore, the creation of salt-tolerant varieties has been widely concerned around the world. It is clear that exposure of seed to salt stress leads to increased accumulation of Na^+^ and Cl^−^ in hypocotyl and endosperm, thereby limiting seed germination ability. Notably, one recent study demonstrated that overexpression of the Arabidopsis *MAN3* gene led to enhanced Cd tolerance due to the increased mannose content in cells [17]. As mentioned above, the mannanase such as MAN5 catalyzes the degradation of mannan backbone in cell wall including endosperm cell wall, thereby releasing mannose units. Together, as members of the same family, it is not difficult to speculate that *MAN5* overexpression has the same effect as *MAN3* overexpression, i.e., they both promote salt stress resistance attributed to their metabolizable mannan as mannose.

As has long been demonstrated, the plant hormone ABA is an inhibitor of seed germination [4]. In contrast to GA, ABA significantly inhibits the expression of endosperm-specific genes [31]. More interestingly, ABI5 has long been shown to function as a crucial factor involved in the feedback regulation of the core ABA pathway, for example, in the regulation of seed germination [32,33]. In addition, ABI5 was found to be mostly active in seeds and prevents germination and post-germinative growth under normal and unfavorable conditions [32]. Notably, the data based on our current study suggests that ABI5 might be involved in negative regulation of *MAN5* gene expression that repressed Arabidopsis seed germination. Consistent with this speculation, we found that the expression of *MAN5* gene in the *abi5* mutant seeds increased faster than in WT seeds. Moreover, since cyanide pretreatment promoted the expression of *MAN5* gene and inhibited the expression of *ABI5* gene, it clearly indicated that *ABI5* is one of the key factors involved in the regulation of *MAN5* expression by cyanide. However, the molecular mechanism of cyanide regulating *ABI5* needs to be further studied. Secondly, the key cis-acting components of *MAN5* gene response to ABI5 regulation also need to be thoroughly studied.

## Figures and Tables

**Figure 1 genes-14-01361-f001:**
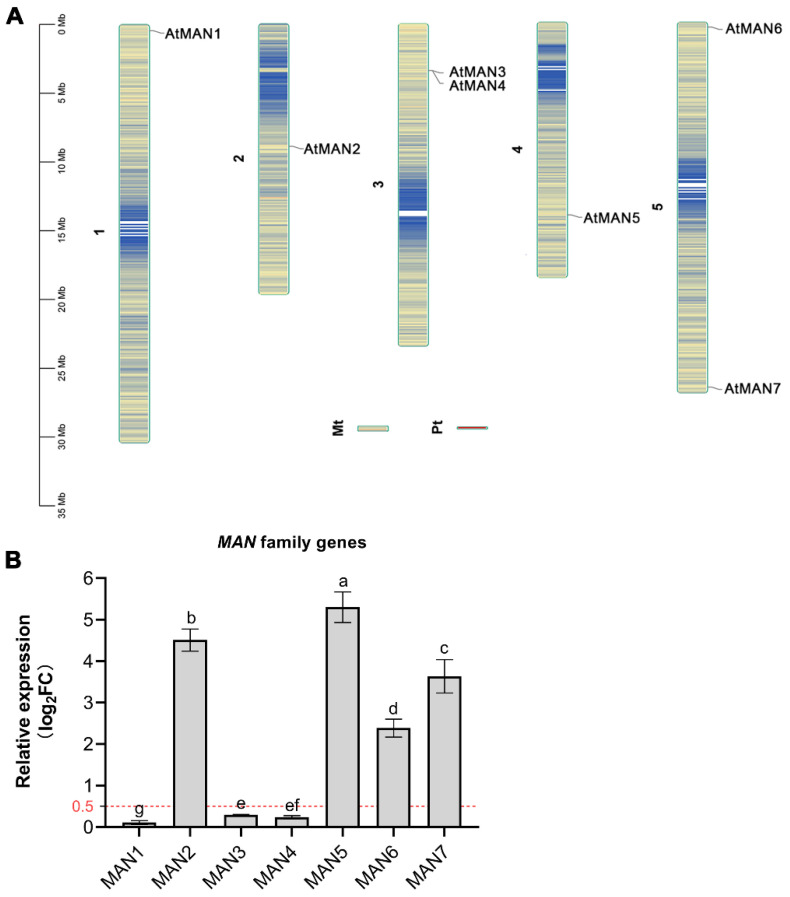
(**A**) Chromosome location of Arabidopsis *MAN* family genes. The chromosomal mapping was drawn by TBtools. (**B**) Changes in relative expression fold of *MAN* genes after cyanide pretreatment compared with untreated controls. Log_2_FC = log_2_foldchange. Significant differences in fold changes of gene expression between different *MAN* genes are denoted by different lowercase letters (*p* < 0.05).

**Figure 2 genes-14-01361-f002:**
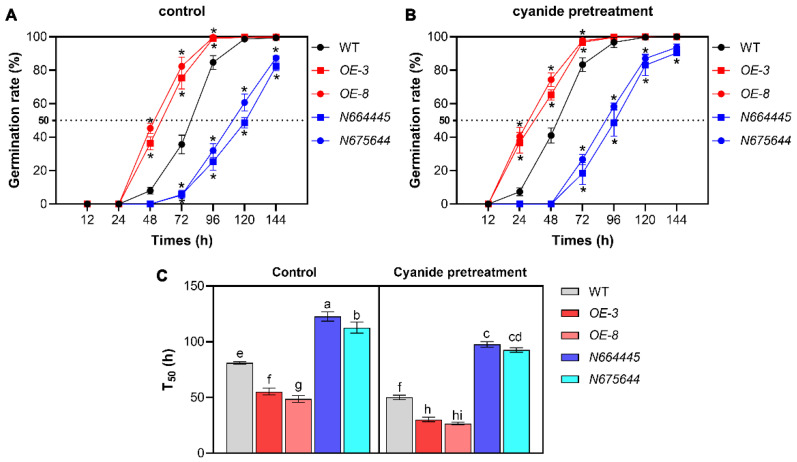
Comparison of seed germination between WT, *MAN5*-OE, and mutants. For this experiment, seeds were pretreated with 10 µmol/L HCN for 12 h and then were sown on 1/2 MS medium for germination assay. Seeds not pretreated with cyanide were not used as controls. Germination rate was detected and compared between WT, *MAN5*-OE, and mutant seeds without (**A**) or with (**B**) cyanide pretreatment. (**C**) T_50_ was compared between WT, *MAN5*-OE, and mutant seeds with or without cyanide pretreatment. T_50_, time to obtain 50% of germinated seeds. Data are the means ± SD of three independent experiments. An asterisk indicates a significant difference in germination between *MAN5* overexpression or mutant seeds and WT seeds (* *p* < 0.05). Significant differences in seed germination rate (T_50_) between different samples are denoted by different lowercase letters (*p* < 0.05).

**Figure 3 genes-14-01361-f003:**
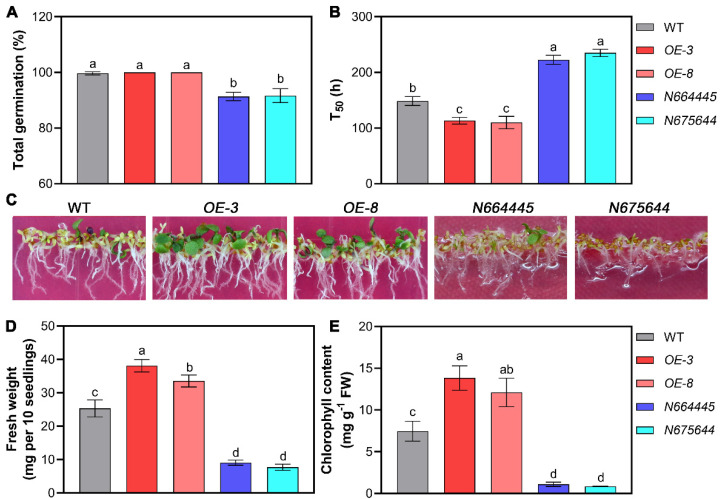
Comparison of seed germination between WT, *MAN5*-OE, and mutants under salt stress conditions. (**A**) Total germination rate and (**B**) T_50_ were compared between WT, *MAN5*-OE, and mutants under the condition of 100 mM NaCl. (**C**) Comparison of the growth of WT, *MAN5*-OE, and mutants after salt stress treatment for 14 days. (**D**) Fresh weight and (**E**) chlorophyll contents were compared between WT, *MAN5*-OE, and mutants after salt stress treatment for 14 days. For fresh weight comparison, 10 seedlings of each type of sample were weighed and analyzed. Data are the means ± SD of three independent experiments. Significant differences between different samples are denoted by different lowercase letters (*p* < 0.05).

**Figure 4 genes-14-01361-f004:**
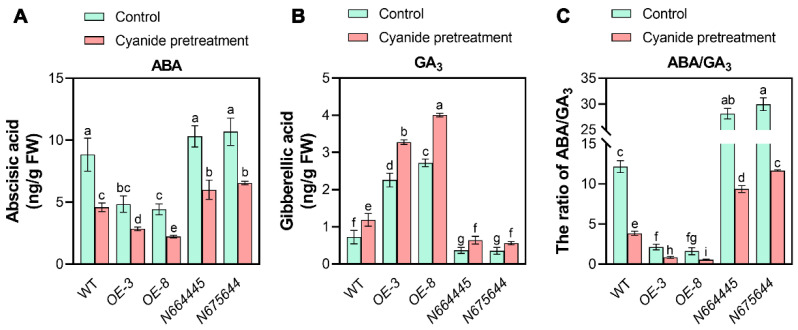
Comparison of ABA and GA_3_ content between WT, *MAN5*-OE, and mutants. For this experiment, Arabidopsis seeds were incubated under dark condition at 25 °C for 2 days with or without cyanide pretreatment, then the contents of (**A**) ABA and (**B**) GA_3_, and (**C**) the ratio of ABA/GA_3_ were detected and compared. Data are the means ± SD of three independent experiments. Significant differences between different samples are denoted by different lowercase letters (*p* < 0.05).

**Figure 5 genes-14-01361-f005:**
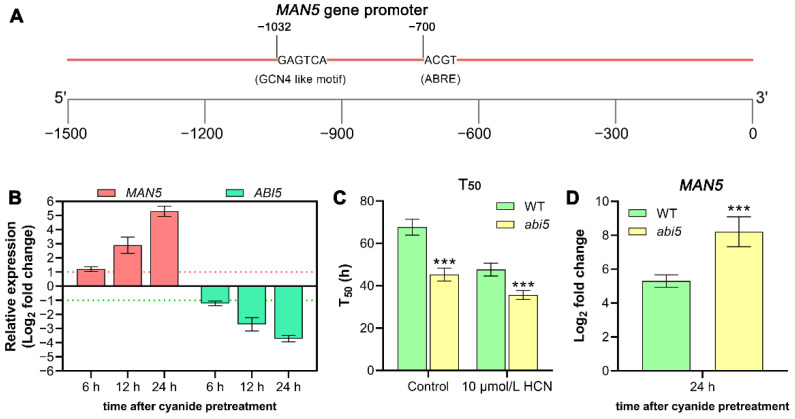
Promoter analysis of the *MAN5* gene. (**A**) Analysis of the cis-acting element of *MAN5* gene promoter. (**B**) Comparative analysis of *MAN5* and *ABI5* gene expression levels by qRT-PCR after cyanide pretreatment for 6, 12, and 24 h. (**C**) Comparison of germination rate (T_50_) between WT and *abi5* mutant seeds with or without cyanide pretreatment. (**D**) Effects of *abi5* mutation on cyanide-induced expression of the *MAN5* gene. Data are the means ± SD of three independent experiments. Significant differences between different samples are denoted by asterisk (*** *p* < 0.001).

## Data Availability

The data that support the findings of this study are available from the corresponding author upon reasonable request.

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
