# Peer review of "MAN5, a Glycosyl Hydrolase Superfamily Protein, Is a Key Factor Involved in Cyanide-Promoted Seed Germination in Arabidopsis thaliana"

_genes, 2023, doi:10.3390/genes14071361_

Round 1
Reviewer 1 Report
This is a clearly written manuscript addressing an interesting question, the basis for cyanide-promotion of seed germination. The authors performed a clear set of experiments testing the connection between HCN treatment, induction of mannanase enzyme, and the role of mannanase in germination using overexpressing and mutated versions of the MAN5 gene in Arabidopsis. Of greatest concern, however, is that data supporting claims regarding effects of cyanide, a key premise for the work, are not directly provided as indicated below:
1. Section 3.2. Several sentences in this section (line 181, 183-84, 186-87) refer to the enhancing effect of HCN on germination. However, the data presented in Fig. 2 do not show the effect of cyanide. The figure shows the effect of the over-expressors and mutants relative to WT MAN5 plants, but does not provide direct statistical comparisons between the samples with and without cyanide. Were the treatments with and without cyanide performed in parallel as part of the same experiment so that statistical comparisons can be made? If so, the relevant statistical information should be included.
2. line 235-238. No data are provided for the claims regarding cyanide pretreatment on GA and ABA. Data should be provided, or the sentences removed.
Other notes:
3. line 83. Please provide reference for floral dip method.
4. line 96. How was a biological rep defined, i.e., how many seeds or plants per rep?
5. line 117. How were the samples ground? Were they frozen? Ground on ice?
6. line 160 refers to chromosomal mapping. Was mapping done as part of this work? If so, add information about how mapping was performed. Alternatively, if gene locations were based on genomic information and annotation obtained from a database (e.g., TAIR), the source of genomic information should be cited.
7. line 163. The primers for the different MAN genes used should be provided in methods (can be in a supplemental table)
8. It would be helpful to the reader to provide rationale for including salt stress in these experiments. There is a fair bit of literature regarding the role of mannose/mannitol and salt stress that may be worth considering.
Minor grammar/word use notes:
1. line 125, 132. ‘filtrated’ should be ‘filtered’
2. line 214. ‘have not expanded’ should be ‘did not expand’
3. line 296. ‘as we all know’ would be better stated as ‘as has long been demonstrated’
Reviewer 2 Report
I find this article interesting in the study of Arabidopsis MANs and their effect on seed germination.
What is missing is more detail in the English of the text, in some parts it is confusing. Also, the discussion should be worked in a better way.
Graph 1b I would like to see the comparison with the untreated ones, the statistical analysis, is it with respect to the untreated ones? the analysis is independent by MAN?
Figure 1, consists of a, b, and c. Where C is not explained in the text, also the model is made by Pymol, what tempering is used? what is the purpose?
Round 2
Reviewer 1 Report
1. Section 3.2. I am afraid that I did not make the concern regarding lack of direct statistical test of the effect of cyanide clear. Fig 2 still does not provide a statistical comparison between the samples with and without cyanide. They only provide a statistical comparison between different mutants and not the effect of cyanide. For example, as shown, it would indicate that WT with and without cyanide is the same (both are labeled ‘c’). Were the treatments with and without cyanide performed in parallel as part of the same experiment so that statistical comparisons can be made? If so, the relevant statistical information needs to be included. if direct comparisons are not possible, claims cannot be made regarding effect of HCN treatment.
2. Section 3.4. Thank you for providing the additional data for the samples with and without cyanide. However, the same concern refers to the effect of cyanide on ABA (Fig 4). The only statistical comparisons made are among genotypes with or without cyanide. If a claim is to be made about the effect of cyanide the comparison must be the same genotype plus/minus cyanide.
Other comments
1.line 195-199. It is not clear why these sentences were added, but they are troubling and do not reflect a scientific approach. There is no reason to make a connection between the proposed 3d structure of the protein resembling a germinating Arabidopsis seed and performing germination experiments.
2. other concerns noted in the prior review have been addressed.
